# Regioselective hydroamination of unactivated olefins with diazirines as a diversifiable nitrogen source

Qingyu Xing[1], Preeti P. Chandrachud[2], Khalilia Tillett[1] & Justin M. Lopchuk ®[1,2,3] ✉

Nitrogen-containing compounds, such as amines, hydrazines, and heterocycles, play an indispensable role in medicine, agriculture, and materials. Alkylated derivatives of these compounds, especially in sterically congested environments, remain a challenge to prepare. Here we report a versatile method for the regioselective hydroamination of readily available unactivated olefins with diazirines. Over fifty examples are reported, including the protecting group-free amination of fourteen different natural products. A broad functional group tolerance includes alcohols, ketones, aldehydes, and epoxides. The proximate products of these reactions are diaziridines, which, under mild conditions, are converted to primary amines, hydrazines, and heterocycles. Five target- and diversity-oriented syntheses of pharmaceutical compounds are shown, along with the preparation of a bis-[15]N diazirine validated in the late-stage isotopic labeling of an RNA splicing modulator candidate. In this work, we report using diazirine (**1**) as an electrophilic nitrogen source in a regioselective hydroamination reaction, and the diversification of the resulting diaziridines.

Amines, hydrazines, and their derivatives persist as pivotal heteroatomic species in medicinal chemistry and drug discovery, playing an indispensable role in the development of novel therapeutic agents[1,2]. Consequently, chemists dedicate a significant portion of their time and effort to the synthesis, functionalization, and purification of nitrogen-containing compounds. Alkyl amines play a crucial role across a spectrum of disease states, including for the treatment of pain, allergies, inflammation, lipid disorders, and cancer (Fig. 1A)[2]. The syntheses of these compounds can pose significant challenges, particularly in sterically congested environments or when pursuing diversity-oriented routes. While archetypical C–N bond forming reactions such as alkylations, cross-couplings, and reductive aminations dominate the toolkit of most medicinal chemistry programs, the escalating demand for rapid access into more diverse and sp³-rich chemical space necessitates the development of new reactions, especially

those that allow for non-traditional retrosynthetic disconnections and the incorporation of electrophilic amine sources.

In principle, a hydroamination reaction addresses this problem by leveraging the ubiquity, structural variety, and commercial availability of olefins in conjunction with reactive nitrogen species in order to produce alkyl amines[3–5]. However, numerous challenges exist in the development of intermolecular versions of these reactions: overcoming the activation barrier of remote olefins with good chemo- and regioselectivity, driving the reaction to completion while avoiding a large excess of either the olefin or amination reagent, and further manipulation of potentially undesired functionality retained on the nitrogen of the resulting products. In recent years, radical-based hydroaminations have become a popular way to circumvent several of these challenges[6,7]. Since the radical intermediates are relatively high in energy, the activation barrier can be overcome, allowing for milder reaction conditions, along with better selectivity and functional group

[1]Department of Chemistry, University of South Florida, Tampa, FL 33620, USA. [2]Drug Discovery Department, H. Lee Moffitt Cancer Center and Research Institute, 12902 Magnolia Drive, Tampa, FL 33612, USA. [3]Department of Oncologic Sciences, College of Medicine, University of South Florida, Tampa, FL 33612, USA. ✉e-mail: Justin.Lopchuk@Moffitt.org

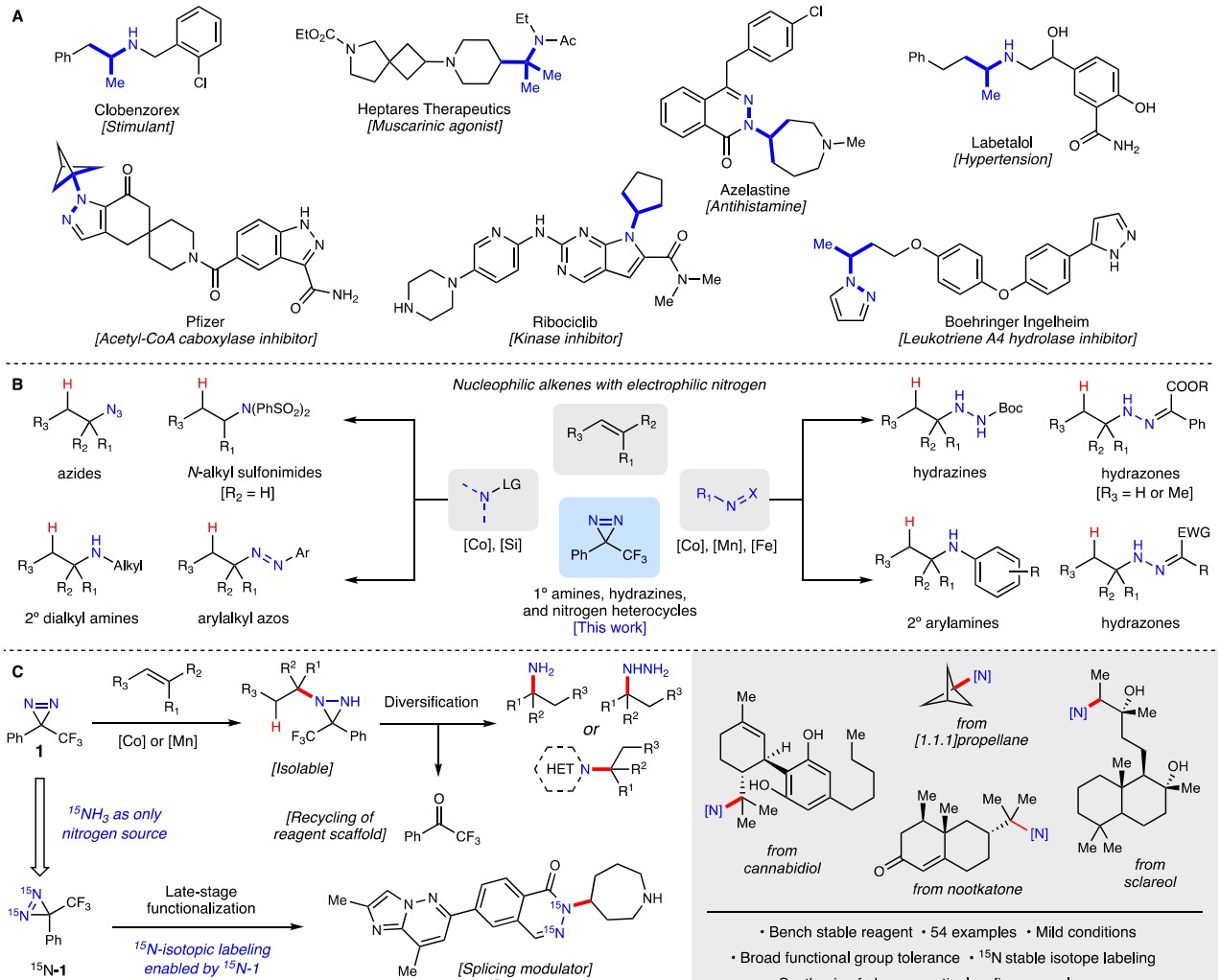

**Fig. 1 | Nitrogen-containing drugs and strategies for their syntheses.**
**A** Pharmaceuticals with a common alkylamine motif. **B** Radical hydroamination approaches via metal-hydride hydrogen atom transfer. **C** This work. Radical hydroamination of unactivated olefins with diazirines as a diversifiable nitrogen donor and its synthetic applications.

tolerance. Nevertheless, the search for a more broadly applicable, diversifiable nitrogen source remains a critical need.

The electrophilic nitrogen sources currently used in radical hydroaminations fall into one of two categories: sp2 or sp3 hybridized nitrogens bearing a leaving group, or sp2 hybridized nitrogens bonded to another nitrogen or oxygen (Fig. 1B). The former group contains reagents such as sulfonyl azides[8,9] that expel the sulfonyl group to give alkyl azides and N-fluorobenzenesulfonimide (NFSI)[10,11], resulting in alkylated benzenesulfonimides. Alkyl azides have been reported for the synthesis of secondary alkyl amines with the loss of molecular nitrogen[12,13]. Arylalkyl azo species can be prepared from tosyl diazenes, with the tosyl serving as the leaving group[14]. The latter group of reagents, where the N=X bond accepts a carbon radical, are exemplified by azodicarboxylates[8,15] (to form protected hydrazines), and electron-deficient diazo compounds[16,17] (to form hydrazones). Aromatic nitro groups are also effective nitrogen sources, delivering hindered secondary anilines[18,19]. While each of these approaches offers a distinct pathway into alkyl-substituted amines via C–N bond formation, all bear extra functionality that may or may not be desirable in the final products making further diversification challenging. Alternatively, diazirines have been shown to engage in both radical and electrophilic amination processes to afford diaziridines that serve as isolable masked amines and hydrazines[20,21]. They are readily cleaved

under mild conditions to the corresponding free amines or hydrazines with concomitant recovery of the ketone backbone, which is then recycled into the diazirine reagent synthesis. The diaziridines can also be used directly in a variety of one-pot/telescoped syntheses of heterocycles (e.g. azoles, pyrroles, aziridines, phthalazinones), obviating the need for the often-troublesome purification of highly polar free amines and hydrazines. Here we report the practical, regioselective hydroamination of unactivated olefins with diazirines (Fig. 1C). Over fifty examples demonstrate the broad scope and functional group tolerance of this reaction. Synthetic applications include target- and diversity-oriented syntheses of numerous pharmaceuticals and clinical candidates. Finally, we report the synthesis of bis-15N diazirine **1**, using 15NH3 as the sole source of the isotopic label, and its deployment in the synthesis of a 15N-labeled RNA splicing modulator.

## Results
### Reaction development and optimization
Our initial forays into the development of diversifiable amination reagents centered around the use of diazirines, which are historically known for their utility as carbene precursors in chemical biology[22,23]. Instead, we recently demonstrated their use in transition metal-catalyzed and photochemical decarboxylative aminations. While many carboxylic acids and their derivatives are commercial and widely

available, this is far less true of the tertiary acids that would enable access to the alkyl amines required for the pharmaceuticals displayed in Fig. 1A. Furthermore, synthetic access into these tertiary acids, especially when remote from other functional groups, is not straightforward. To overcome this hurdle, we turned our attention to the use of olefins in concert with diazirine 1. Substituted olefins are abundantly available, in both commercial building blocks and natural products, and are simple to prepare in the laboratory via robust chemical processes (e.g. alkylation, cross- or ring-closing metathesis, Wittig, etc.).

Carreira's elegant hydrohydrazination work served as inspiration for the immediate starting point due to the mild cobalt-catalyzed conditions and good regioselectivity that were reported[8]. Indeed, upon initial reaction of diazirine 1 with 4-phenylbut-1-ene, low yields of the corresponding diaziridine were observed. While this served as confirmation that the C−N bond formed, extensive optimization of solvent, catalyst, and hydride source proved ineffective for raising the yield. However, inspired by Nojima, di-t-butyl peroxide was added to facilitate the formation of the cobalt hydride[24]. To our delight, this proved the key to unlocking the reaction (see Supplementary Fig. 9 for proposed mechanism and catalytic cycle). After a switch to olefin 2 as our model substrate (lower volatility compared to 4-phenylbut-1-ene), a re-optimization of catalyst, solvent, and hydride source afforded diaziridine 3a in 99% isolated yield (Table 1).

Other cobalt catalysts such as Co(dpm)$_3$ (entry 1), and Co(TPP)Cl (entry 2) gave trace amounts of the desired product. Mn(dpm)$_3$ (entry 3), failed to show any appreciable regioselectivity (ca. 1.5:1), however, the products were obtained in 86% isolated yield after only 2 h at 0 °C. Ultimately, this turned out to be an excellent protocol for symmetrical olefins and will be discussed further (see below). Other counterions for the salen catalyst (e.g. OAc in cat-2, Cl in cat-3, entries 4 and 5), led to diminished yields as did Carreira's catalyst (cat-4, entry 6). While a mixed solvent system (DCE:IPA 4:1) was found to be optimal for most substrates, IPA alone (entry 7) gave 3a in a slightly lower yield and longer reaction time (30 h). The IPA only conditions proved to be useful with tri-substituted alkenes as the substrate scope was further evaluated. Although the olefin was consumed within 3 h, a longer reaction time was required to achieve completion. At 3 h the isolated yield was 66% (entry 8), whereas prolonging the reaction time to 16 h increased the isolated yield to 93% (entry 9) and to 20 h, 99% (optimized conditions). As mentioned above, di-t-butyl peroxide was a crucial additive; in its absence, the yield dropped to 38% (entry 10). When using t-butyl peroxide instead, the yield decreased to 36% (entry 11), and the Mukaiyama hydration product was isolated instead as the major product. Other silanes, such as TES and PHMS, gave only trace amounts of 3a (entries 12 and 13). Higher temperatures were deleterious to the yield (60 °C, 56% yield, entry 14) as was running the reaction with no precautions (open to air and light, 54% yield, entry 15).

## Table 1 | Optimization of the reaction

| Entry[a] | Deviation from above | Yield (%)[b] |
|---|---|---|
| 1 | Co(dmp)$_3$ | trace |
| 2 | Co(TPP)Cl | trace |
| 3 | Mn(dmp)$_3$ | 51/35[c,d] |
| 4 | cat-2 | 51 |
| 5 | cat-3 | 39 |
| 6 | cat-4 | 57 |
| 7 | IPA only | 84[e] |
| 8 | 3 h | 66 |
| 9 | 16 h | 93 |
| 10 | w/o t-BuOOt-Bu | 38 |
| 11 | t-BuOOH (2.5 eq)[f] | 36 |
| 12 | TES | trace |
| 13 | PHMS | trace |
| 14 | 60 °C | 56 |
| 15 | no precautions[g] | 54 |

[a]Reactions were conducted with 2 (0.1 mmol), 1 (0.15 mmol, 1.5 eq), PhSiH$_3$ (0.1 mmol, 1 eq), cat-1 (0.005 mmol, 0.05 eq), DCE (400 μL) and IPA (100 μL), 40 °C for 20 h under argon atmosphere.
[b]Isolated yield.
[c]Reaction conducted with 2 (0.1 mmol), 1 (0.15 mmol, 1.5 eq), PhSiH$_3$ (0.1 mmol, 1 eq), Mn(dpm)$_3$ (0.005 mmol, 0.05 eq), DCE (400 μL) and IPA (100 μL), 0 °C for 2 h under argon atmosphere.
[d]Markovnikov/anti-Markovnikov ratio.
[e]Reaction run for 30 h.
[f]5.5 M solution in decane (dried over 4 Å molecular sieves).
[g]Reaction run under air atmosphere with no protection from light. Co(dpm)$_3$ = tris(2,2,6,6-tetramethyl-3,5-heptanedionato)cobalt(III). Co (TPP)Cl = 5,10,15,20-tetraphenyl-21H,23H-porphine cobalt(III) chloride. Mn(dpm)$_3$ = tris(2,2,6,6-tetramethyl-3,5-heptanedionato)manganese(III). TES = triethylsilane, PHMS = polymethylhydrosiloxane. DCE = dichloroethane. IPA = isopropyl alcohol.

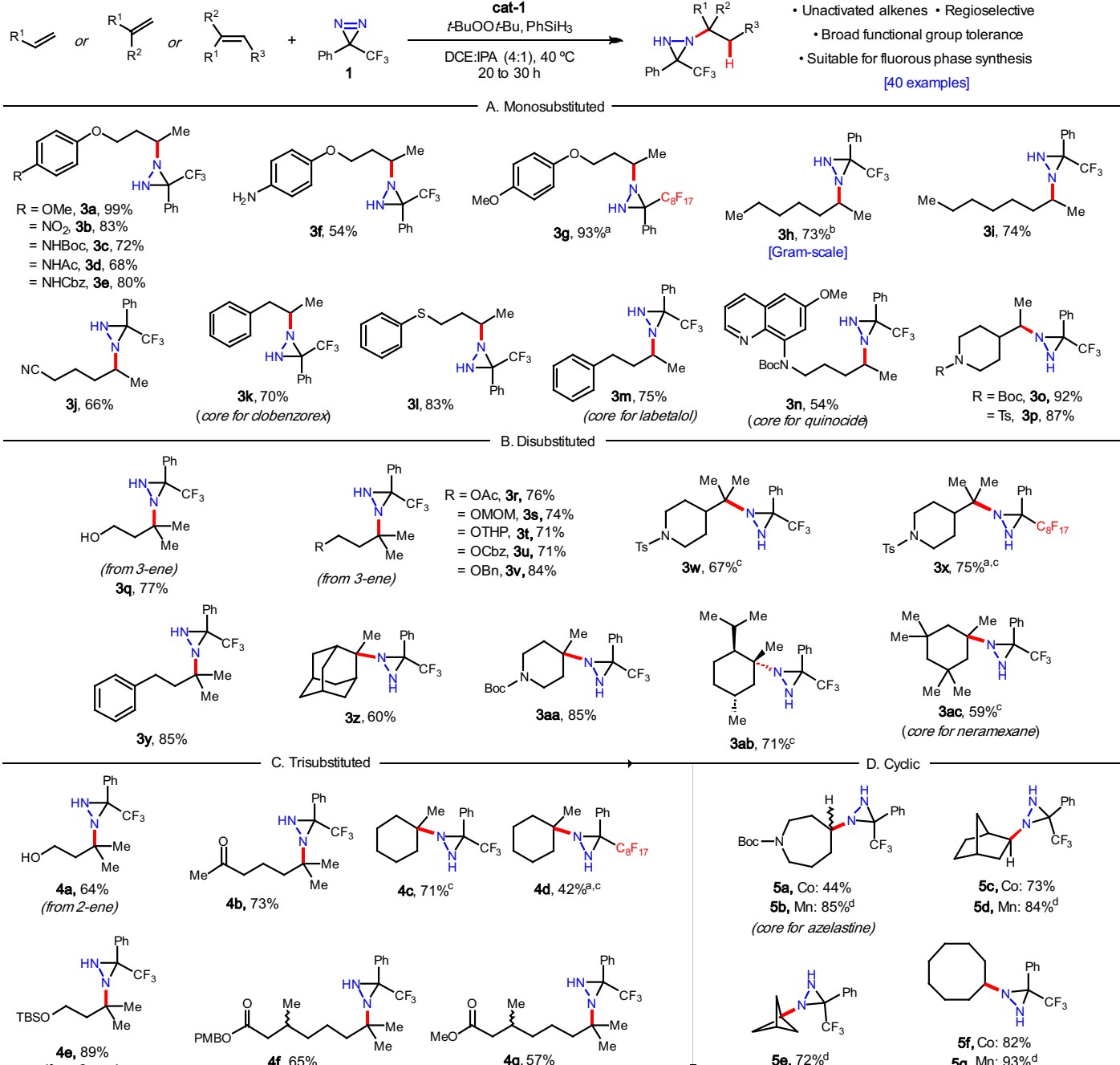

**Fig. 2 | Scope of the hydroamination - building blocks and pharmaceutical core structures. A** Monosubstituted alkene examples. **B** Disubstituted alkene examples. **C** Trisubstituted alkene examples. **D** Cyclic alkene examples. Reactions conducted with alkene (0.1 mmol), **1** (0.15 mmol, 1.5 eq), cat-1 (0.005 mmol, 0.05 eq), t-BuOOt-Bu (0.1 mmol, 1 eq), PhSiH₃ (0.1 mmol, 1 eq), DCE (400 µL) and IPA (100 µL) at 40 °C for 20 h under argon atmosphere with protection from light. Isolated yields are reported. ᵃReaction run at 0.1 mmol scale with perfluorinated diazirine **1**.

ᵇReaction run at 1 g scale. ᶜReaction run with IPA (100 µL) as the only solvent for 30 h. ᵈReaction conducted with alkene (0.1 mmol), **1** (0.15 mmol, 1.5 eq), Mn(dpm)₃ (0.005 mmol, 0.05 eq), PhSiH₃ (0.1 mmol, 1 eq), DCE (400 µL) and IPA (100 µL) at 0 °C for 2 h under argon atmosphere with protection from light.
Mn(dpm)₃ = Tris(2,2,6,6-tetramethyl-3,5-heptanedionato)manganese(III). DCE = dichloroethane, IPA = isopropyl alcohol.

## Substrate scope

With the optimized conditions in hand, the scope of building blocks containing unactivated olefins was broadly evaluated. Monosubstituted olefins (Fig. 2A), cyclic or acyclic disubstituted olefins (Fig. 2B), cyclic or acyclic trisubstituted olefins (Fig. 2C), and *cis*-cyclic olefins (Fig. 2D) all performed well, affording the desired diaziridine products as single regioisomers in good to excellent yields. Nitroarenes (**3b**) and classical aryl and aliphatic amine protecting groups were tested (e.g. Boc (**3c, 3n, 3o, 3aa and 5a/b**), Ac (**3d**), Cbz

(**3e**) and Ts (**3p, 3w** and **3x**); all were found to be compatible with the reaction conditions and, importantly, offer the opportunity for downstream synthetic manipulations of the orthogonally protected diamine motifs. In a similar vein, a free amine was tolerated (**3f**, 54% yield), and the presence of a nitrile group (**3j**) was also supported. Free hydroxy groups (**3q, 4a**) were compatible, as were many of their common protecting groups including acetyl (**3r**), MOM (**3s**), THP (**3t**), Cbz (**3u**), Bn (**3v**) and TBS (**4e**). Phenolic (**3a-f, 3g**) and thiophenolic ethers (**3l**), ketones (**4b**) and esters (**4f** and **4g**) could be

**Fig. 3 | Scope of the hydroamination - terpene natural products.** Reactions conducted with alkene (0.1 mmol), **1** (0.15 mmol, 1.5 eq), cat-1 (0.005 mmol, 0.05 eq), t-BuOOt-Bu (0.1 mmol, 1 eq), PhSiH₃ (0.1 mmol, 1 eq), DCE (400 μL) and IPA (100 μL) at 40 °C for 20 h under argon atmosphere with protection from light. Isolated yields are reported. ᵃReaction run with IPA (500 μL) as only solvent for 30 h. DCE dichloroethane, IPA isopropyl alcohol.

incorporated affording the desired products in good to excellent yields.

Various simple hydrocarbons were successfully employed: linear alkenes (**3h, 3i**), methylenecyclohexane derivatives (**3ab, 3ac**) phenyl derivatives (**3k, 3m, 3y**), an adamantyl derivative (**3z**), methyl cyclohexenes (**4c, 4d**), norbornene (**5c/d**), and cyclooctene (**5f/g**). Heterocyclic substrates included Boc- or Ts-protected piperidines (**3o, 3p, 3w, 3x, 3aa**), an azepine (**5a/b**), and an 8-amino quinoline (**3n**). This last example is notable since both the starting material and diaziridine product may act as a ligand with the catalyst, yet **3n** is still obtained in 54% isolated yield. *Cis*-olefins were generally hydroaminated in good yields (**5c, 5f**), however with azepine derivative **5a**, a lower isolated yield was obtained (44%) due to loss of the Boc protecting group. This appears unique to this example since this phenomenon was not observed with other Boc-protected substrates (e.g. **3o, 3n, 3aa**). While the cobalt-catalyzed protocol worked well in these examples, for symmetrical alkenes (Fig. 2D) the manganese-catalyzed version proved superior, giving the desired products in shorter reaction times with better yields (**5b, 5d, 5g**). To our delight, [1.1.1]propellane engaged well with the alternative manganese conditions, delivering **5e** in 72% yield, which affords a different pathway into amine, hydrazine, and heterocyclic bicyclopentyl building blocks[25].

The reaction was conducted on gram-scale where **3h** was isolated in 73% yield. Previously we demonstrated the use of a perfluorinated diazirine reagent (with C₈F₁₇) that allowed for both the initial diazirine reaction and subsequent diaziridine cleavage to be incorporated into fluorous phase workflows[20]. This eliminates the need for column chromatography and expedites the library synthesis of nitrogen-containing compounds for discovery scientists. Monosubstituted olefins (**3g**), disubstituted olefins (**3x**), and trisubstituted olefins (**4d**) all gave the expected hydroamination products, though **4d** suffered a loss in yield. Many of the olefins displayed in Fig. 2 can be readily

crafted into pharmaceuticals through known transformations including clobenzorex (from **3k**), labetalol (from **3m**), neramexane (from **3ac**), quinocide (from **3n**) and azelastine (from **5a/b**).

Having evaluated the initial scope and functional group compatibility of the hydroamination we turned our attention to the late-stage functionalization of more complex natural products with an emphasis on terpenoids (Fig. 3). Terpenes and terpenoids, many of which are commercially available, are not only common in the food and fragrance industries, but also serve as effective building blocks in medicinal chemistry and natural product synthesis. While these scaffolds appear sporadically throughout the hydrofunctionalization literature, to date there is no general hydroamination that has been demonstrated to be effective across numerous members of the class. Classical methods reported for C–N bond formation on terpenes include the Ritter reaction[26], mercuration with anilines and azides[27,28], and others. More recent hydrofunctionalization methods include Boger's hydroazidation of citronellol[29], Carreira's hydrohydrazination of camphene[8], Glorius' iminative bis-functionalization of perillol, limonene, and nootkatone[30], Lin and Xu's independent approaches to the electrochemical diazidation of limonene oxide and nootkatone[31,32], and Engle's aminoarylation of sclareol[33]. Monosubstituted olefins that successfully underwent the diazirine-based hydroamination include eugenol (**6a**) and sclareol (**6i**). Reaction on the disubstituted isopropenyl group of nootkatone (**6b**), cannabidiol (**6c**), perillol (**6d**), limonene oxide (**6e**), isopulegol (**6f**), valencene (**6g**), and limonene (**6k**) all afforded the desired products in moderate to good yields. Bisabolol (**6h**), citronellic acid (**6l**), citronellal (**6m**), and citronellol (**6n**) serve as examples of suitable trisubstituted prenyl groups that were able to undergo the hydroamination. Finally, camphene was converted to **6j** selectively as the exo-product in 75% yield.

Importantly, each of the above reactions were performed directly on the natural products, without the need for protecting

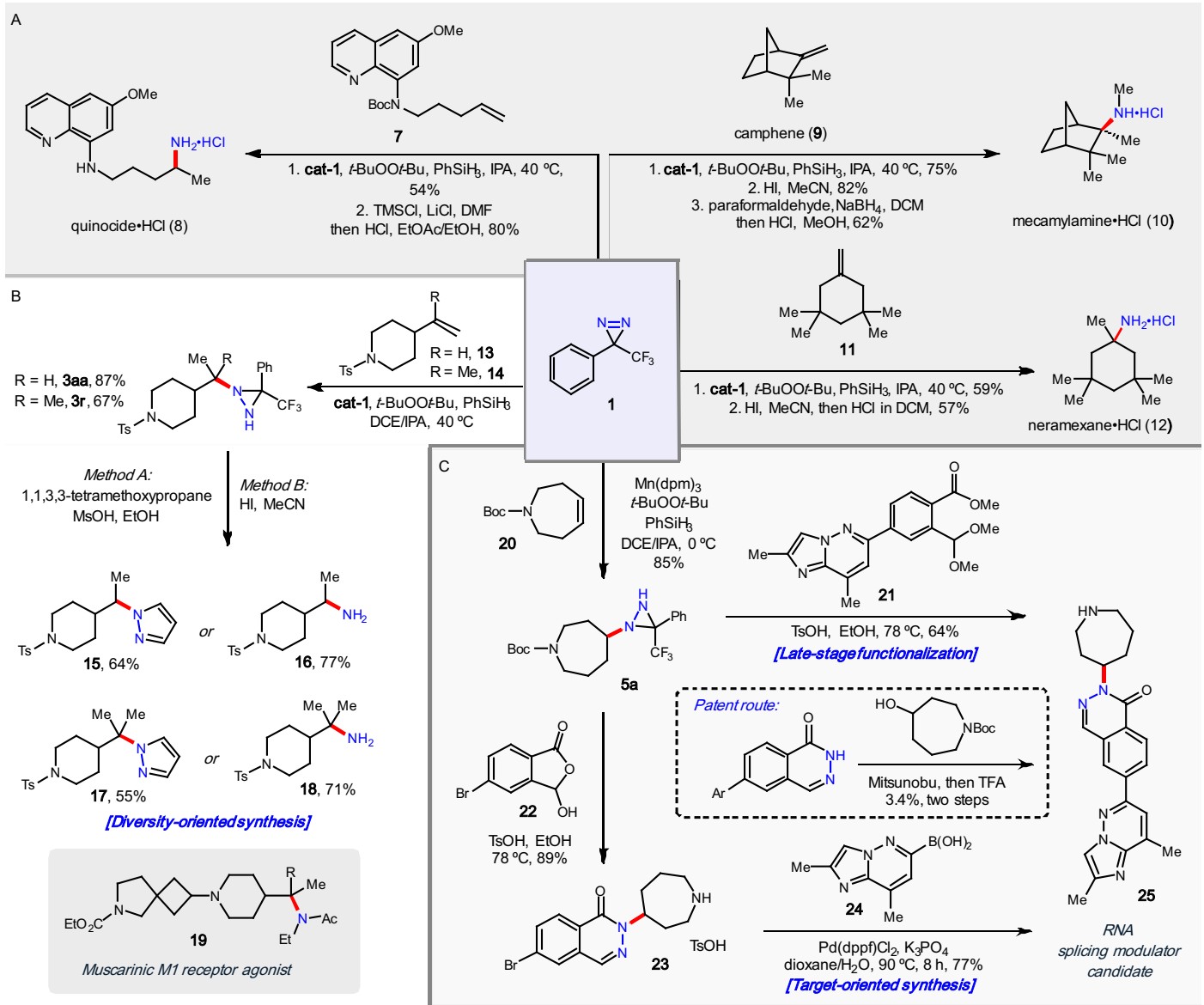

**Fig. 4 | Synthetic applications: Target- and diversity-oriented synthesis of pharmaceuticals and related compounds via hydroamination. A** Target-oriented synthesis of mecamylamine, neramexane, and quinocide. **B** Diversity-oriented synthesis of four piperidine-based fragments of a muscarinic M1 receptor candidate. **C** Target-oriented and late-stage functionalization approaches to a splicing modulator candidate.

groups. As a result, the functional group tolerance was further illuminated: α,β-unsaturated ketones (**6b**), free phenols (**6a, 6c**), allylic alcohols (**6d**), aliphatic alcohols (**6f, 6h, 6i, 6n**), epoxides (**6e**), carboxylic acids (**6l**) and aldehydes (**6m**) were all tolerated. Notably, kinetically-driven chemoselectivity was observed whereby disubstituted olefins reacted significantly faster than trisubstituted olefins. Thus, for cannabidiol (**6c**), perillol (**6d**), valencene (**6g**), bisabolol (**6h**), and limonene (**6k**) the observed products were a result of mono-hydroamination only on the isopropenyl group, leaving the trisubstituted olefins untouched. However, when the reaction time was increased to 30 h in the presence of a twofold amount of diazirine, catalyst, peroxide, and silane, the bis-hydroamination products appeared and slowly became more dominant. Presumably this is a steric effect of the initial binding of the metal and migratory insertion to the olefin. However, in the mono-aminated substrates displayed in Figs. 2 and 3, the steric environment around (but not directly attached to) the reactive olefin appears to have significantly less influence as shown in **3o, 6c**, and **6j**.

## Synthetic applications

With the substrate scope in hand, we endeavored to highlight the utility of the diaziridine intermediates through their conversion to amines, hydrazines, and various heterocyclic species. As shown in Fig. 4, this was achieved with a variety of target-oriented syntheses, diversity-oriented approaches, and late-stage functionalization, all from diazirine reagent **1**. Primaquine, a member of the 8-aminoquinoline class of drugs, is used for the prevention and treatment of malaria and is found on the World Health Organization's List of Essential Medicines. Quinocide is the constitutional isomer of primaquine, and the major contaminant formed during its synthesis[34]. Access to quinocide (**8**) is critical for quality control and impurity assessment of primaquine; however, its availability is somewhat limited and at an exorbitant cost. The treatment of quinoline **7** under the cobalt-catalyzed conditions gave diaziridine **3n** 54% yield, which was then exposed to TMSCl and LiCl in DMF to reveal the amine with concomitant cleavage of the Boc group to produce quinocide•HCl (**8**) in 80% yield (Fig. 4A). This compares favorably to the previous route

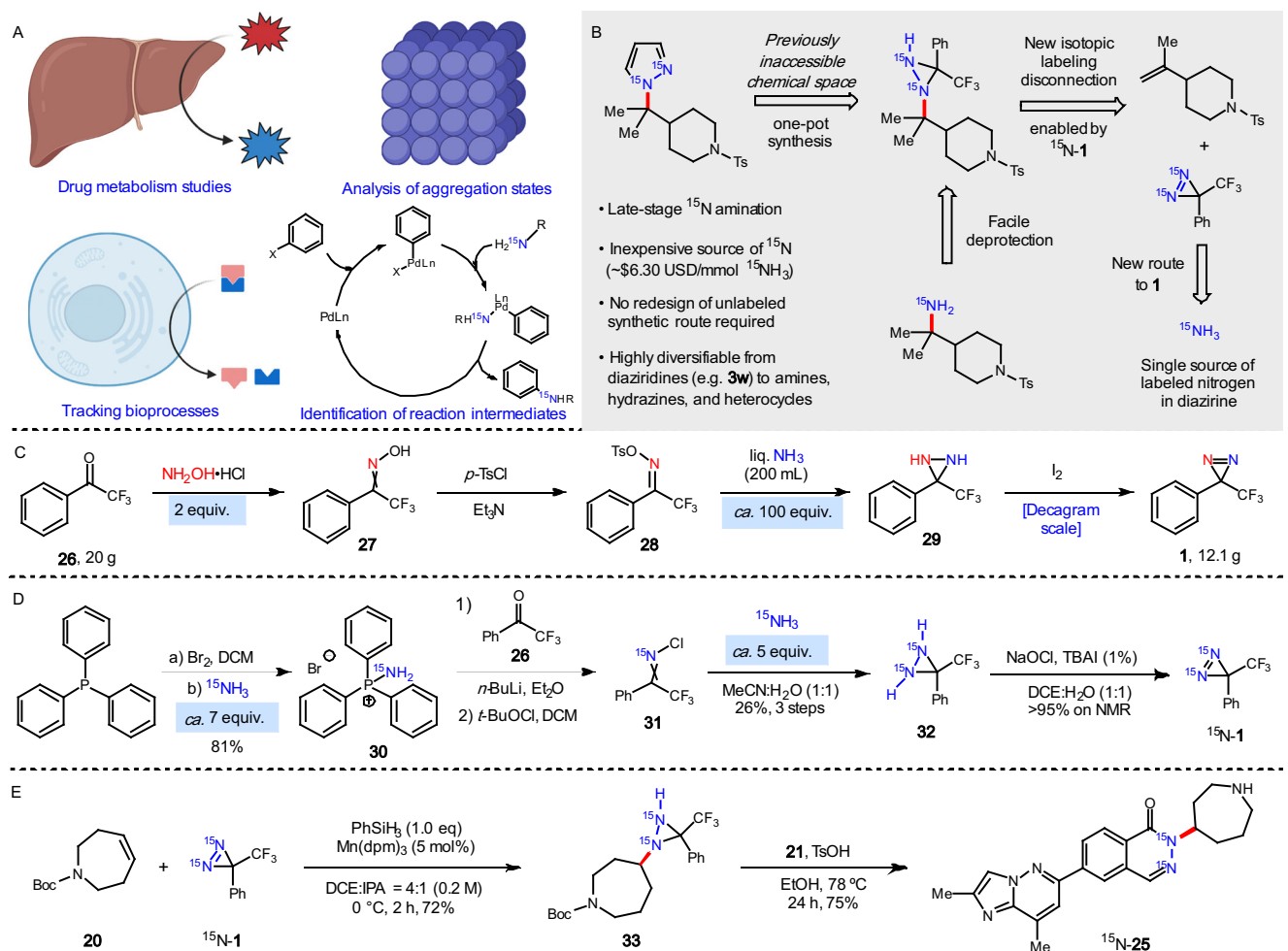

**Fig. 5 | Late-stage $^{15}$N installation and its potential applications. A** Applications of $^{15}$N-containing molecules. **B** Hydroamination with $^{15}$N-diazirine to insert $^{15}$N-atoms into target molecules. **C** Previous unlabeled decagram synthesis of diazirine **1**. **D** Redesigned synthesis of $^{15}$N-diazirine ($^{15}$N-**1**) from $^{15}$N-ammonia as the only nitrogen source. **E** Synthesis of $^{15}$N-containing drug candidate $^{15}$N-**25**. (Panel A Created with BioRender.com released under a Creative Commons Attribution-NonCommercial-NoDerivs 4.0 International license (https://creativecommons.org/licenses/by-nc-nd/4.0/deed.en)).

where the amine was installed via a six-step sequence originating from nitroethane[35]. Mecamylamine is an antagonist of nicotinic acetylcholine receptors that is used for smoking cessation and hypertension[36]. Following the selective hydroamination of camphene (**9**) as previously discussed, diaziridine **6j** was converted to the free amine with HI (82% yield) followed by reductive amination with paraformaldehyde to afford mecamylamine•HCl (**10**) in 62% yield. This provides a more tractable preparatory scale route for discovery, which avoids the use of hydrogen cyanide that is common in the industrial process (Fig. 4A). Neramexane is an NMDA antagonist that is being clinically investigated for a number of indications including Alzheimer's disease and tinnitus[37,38]. The reaction of diazirine **1** with 1,1,3,3-tetramethyl-5-methylenecyclohexane (**11**) produced diaziridine **3ac** in 59% yield, which was hydrolyzed in the presence of HI to furnish neramexane•HCl (**12**) in 57% yield (Fig. 4A).

Synthetic medicinal chemistry workflows are best expedited through common intermediates that can be readily diversified for structure-activity relationship (SAR) studies. Azaspirocycle **19** is a muscarinic M1 receptor agonist from Heptares Therapeutics with potential applications in various neurological disorders[39]. Piperidine building blocks **15, 16,** and **18** were targeted by the researchers via three distinct routes and three different sets of intermediates using classical chemistry: reductive aminations, $S_N2$ displacements, and nucleophilic additions. Piperidine **17**, an obvious derivative to be

examined for SAR in this subset of fragments, could not be prepared through these routes and thus was not evaluated. In a more streamlined approach, piperidines **13** and **14** were subjected to the hydroamination conditions with diazirine **1** to afford diaziridines **3p** and **3w** in 87% and 67% yield, respectively. Treatment with either pyrazole-forming conditions (Method A) or cleavage to the free amine (Method B) rapidly delivered all four building blocks (**15**–**18**) in 55-77% yield (Fig. 4B). Taken together with our previously reported diversification reactions, it is easily envisioned how a small number of strategically chosen intermediates can be multiplied into a large library of medicinally relevant scaffolds covering a significant amount of chemical space.

Phthalazinone **25** is a splicing modulator, developed by Remix Therapeutics, that may be useful in treating disease through targeting RNA[40]. In the original route, it was prepared from 6-bromophthalazin-1(2H)-one over three steps in ca. 3% overall yield. The lowest yielding steps are the Mitsunobu reaction to install the azepine, followed by deprotection (Fig. 4C, inset). By leveraging the manganese-catalyzed version of our hydroamination, symmetrical cyclic olefin **20** was smoothly converted into diaziridine **5b** in 85% yield. Condensation of **5b** with ester **21** effected a late-stage functionalization to afford splicing modulator **25** in 64% yield. This approach can be used not only to rapidly prepare **25** in high yield (54% yield over two steps from olefin **20**), but also interrogate the SAR around the azepine. Alternatively, a

three-step approach can be conducted that would efficiently enable the late-stage variation of heterocycle **25**. Here, diaziridine **5b** was first condensed with 5-bromo-3-hydroxyisobenzofuran-1(3*H*)-one **22** in 89% yield, followed by Suzuki coupling with **24** to afford the desired product in 77% yield (58% yield over three steps from olefin **20**) (Fig. 4C).

### $^{15}$N isotopic labeling

The stable isotopic labeling of small molecules is an important analytical tool used across a variety of fields due to its low natural isotopic abundance (e.g. $^2$H, $^{13}$C, $^{15}$N), which delivers a high signal to noise ratio when observed in NMR, MS, and MRI. Recent advances in newer techniques such as hyperpolarization further increase the sensitivity[41]. The synthesis of $^{15}$N-labeled pharmaceuticals and agrochemicals allows for the in vivo study of both the metabolism and environmental fate of candidate molecules (Fig. 5A)[42]. $^{15}$N NMR has also been used to study protein packing and the conformation of protein-ligand complexes[43]. The incorporation of $^{15}$N monitoring by mass spectrometry enables the quantification of the metabolism of proteins and other biomolecules (e.g. metabolomics)[44]. The elucidation of organic and organometallic mechanisms is also enhanced through the monitoring of reaction intermediates and determination of kinetics[45].

Despite the utility of $^{15}$N-containing molecules, their syntheses still predominantly rely on $^{15}$N-ammonium salts[46], or a small number of simple building blocks such as $^{15}$N-ammonia[47], $^{15}$N-hydroxylamine[47], $^{15}$N-hydrazine monohydrate[48], and $^{15}$N-urea[49]. In nearly all cases, this necessitates a time-consuming radical redesign of the synthetic route for the small molecule that needs to be labeled. Several efforts have been made toward the development of an electrophilic $^{15}$N-nitrogen source. Jones reported a $^{15}$N-diazonium reagent that afforded an azo species en route to the preparation of $^{15}$N-labeled deoxynucleosides[50]. Unkefer developed 1-chloro-1-[$^{15}$N]nitrosocyclohexane for the generation of $^{15}$N-labeled amino acids[51]. While useful for their given targets, both approaches lack generality. Bis-$^{15}$N labeled diazirine (**1**) can serve as a more universal $^{15}$N-labeled electrophilic nitrogen source when coupled not only with the hydroamination chemistry described above, but also with the previously reported decarboxylative aminations. The mild reactions conditions allow for the label(s) to be installed at a late-stage, minimizing cost and waste, and the diversification of the diaziridines affords immediate access to myriad $^{15}$N amines, hydrazines, and nitrogen-containing heterocycles. Most importantly, the tedious redesign of synthetic routes would no longer be required when unlabeled diazirine **1** is used for the initial synthesis (Fig. 5B).

The current synthesis of diazirine **1** requires using hydroxylamine to install the first nitrogen atom and a large excess of liquid ammonia to install the second. This route is neither practical nor economically feasible for the synthesis of $^{15}$N-labeled diazirine **1** with *ca.* 100 eq of $^{15}$N-ammonia going to waste (Fig. 5C). While other methods exist for the conversion of ketones to diazirines, none are suitable for α,α,α-trifluoroacetophenone[47,52]. Instead, we endeavored to devise a new route to $^{15}$N-**1** that used $^{15}$NH$_3$ as the lone source of isotopically labeled nitrogen (Fig. 5D). Toward this end, a modified literature procedure was used to generate $^{15}$N-phosphorus-nitrogen ylide **30**[53], which required *ca.* 7 equivalents of $^{15}$NH$_3$. This bench stable intermediate was treated with n-BuLi and ketone **26** to affect a Wittig-like reaction that furnished the corresponding $^{15}$N-labeled imine. Treatment of the imine with t-BuOCl followed by $^{15}$NH$_3$ (*ca.* 5 equiv.) allowed the oxidation/cyclization sequence to proceed and delivered $^{15}$N-diaziridine **32**. Oxidation with NaOCl and catalytic TBAI gave $^{15}$N-**1** (>95% yield by NMR). To demonstrate the effectiveness of the late-stage isotopic incorporation, cyclic olefin **21** was treated with $^{15}$N-**1** under the manganese-catalyzed hydroamination conditions, which furnished $^{15}$N-labeled diaziridine **33** in 72% yield. Condensation of $^{15}$N-diaziridine **33** with ester **21** delivered splicing modulator $^{15}$N-**25** in 75% yield

(Fig. 5E). The simple reagent synthesis, mild late-stage functionalization conditions, and ability to engage $^{15}$N-**1** under any present or future diazirine protocol makes this an attractive option for stable isotope incorporation.

## Methods

### General procedure A: cobalt catalyzed Markovnikov-type hydroamination

To a flame-dried reaction vial equipped with a magnetic stir-bar and rubber-lined cap under argon atmosphere was added catalyst cat-1 (3.9 mg, 0.005 mmol, 0.05 eq) and the vial back-flushed with argon twice, followed by addition of a mixture of anhydrous DCE: IPA (4:1, 500 μL) via syringe, resulting in a dark green solution. To this solution, alkene (0.100 mmol, 1 eq.), diazirine **1** (28.0 mg, 0.150 mmol, 1.5 eq), t-BuOOt-Bu (14.6 mg, 18.4 μL, 0.100 mmol, 1 eq) and phenylsilane (10.8 mg, 12.3 μL, 0.100 mmol, 1 eq) were added sequentially via syringe. The vial was covered with aluminum foil and stirred at 40 °C for 20 h. The crude reaction mixture was dried *in vacuo*, adsorbed onto silica gel, and purified via flash column chromatography on silica gel.

### General procedure B: cobalt catalyzed Markovnikov-type hydroamination with IPA

To a flame-dried reaction vial equipped with a magnetic stir-bar and rubber-lined cap under argon atmosphere was added catalyst cat-1 (3.9 mg, 0.005 mmol, 0.05 eq) and the vial back-flushed with argon twice, followed by addition of anhydrous IPA (500 μL) via syringe, resulting in a dark green solution. To this solution, alkene (0.100 mmol, 1 eq.), diazirine **1** (28.0 mg, 0.150 mmol, 1.5 eq), t-BuOOt-Bu (14.6 mg, 18.4 μL, 0.100 mmol, 1 eq) and phenylsilane (10.8 mg, 12.3 μL, 0.100 mmol, 1 eq) were added sequentially via syringe. The vial was covered with aluminum foil and stirred at 40 °C for 20 h. The crude reaction mixture was dried *in vacuo*, adsorbed onto silica gel, and purified via flash column chromatography on silica gel.

### General procedure C: Mn-catalyzed hydroamination

To a flame-dried reaction vial equipped with a magnetic stir-bar and rubber-lined cap under argon atmosphere was added Mn(dpm)$_3$ (3.0 mg, 0.005 mmol, 0.05 eq) and the vial back-flushed with argon, followed by addition of a mixture of anhydrous DCE: IPA (4:1, 500 μL) via syringe, resulting in a black suspension. To this mixture, alkene (0.100 mmol, 1 eq.), diazirine **1** (28.0 mg, 0.150 mmol, 1.5 eq), and phenylsilane (10.8 mg, 12.3 μL, 0.100 mmol, 1 eq) were added sequentially via syringe. The reaction vial was then cooled to 0 °C in an ice bath and covered with aluminum foil and stirred for two hours until TLC monitoring indicated total consumption of the starting material. A color change from black to yellow or brown (substrate dependent) also indicated reaction completion. The crude reaction mixture was dried *in vacuo*, adsorbed onto silica gel, and purified via flash column chromatography on silica gel.

## Data availability

The data generated in this study, including NMR, HRMS, and yields, are availability in the paper and its Supplementary Information. Detailed experimental conditions are availability in the Supplementary Information. All data are available from the corresponding author upon request.

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

## Acknowledgements

We gratefully acknowledge the National Science Foundation (CHE-2301063, J.M.L., discovery and scope of the hydroamination) and the National Institutes of Health (NIGMS R35-GM142577, J.M.L., synthetic applications and $^{15}$N labeling) for support of this research. This work has also been supported in part by the Chemical Biology Core Facility at the H. Lee Moffitt Cancer Center & Research Institute, an NCI designated Comprehensive Cancer Center (P30-CA076292). We thank Harshani Lawrence Moffitt for NMR and HRMS support.

## Author contributions

P.P.C and J.M.L. conceived and designed the initial project. Q.X. and J.M.L. conceived and designed the applications and $^{15}$N labeling. Q.X, P.P.C., and K.T. performed the experimental studies. Q.X, P.P.C. K.T., and J.M.L. analyzed and interpreted experimental data. Q.X., K.T., and J.M.L. wrote the manuscript. All authors reviewed, commented on, and approved the final draft.

## Competing interests

The authors declare no competing interests.
