## [Peer Review File · Nature Communications]

Regioselective Hydroamination of Unactivated Olefins with Diazirines as a Diversifiable Nitrogen SourceREVIEWER COMMENTS

Reviewer #1 (Remarks to the Author):

Lopchuk et al. reported a practical, regioselective (Markovnikov) hydroamination and hydrohydrazination of unactivated olefins (mono-, di-, and trisubstituted). This novel strategy involved using trifluoromethyl phenyl diazirines as an electrophilic source of nitrogen with Cobalt or Manganese catalysts. The diaziridines thus obtained were converted into corresponding amines, hydrazines, or directly into N-heterocycles (azoles, pyrroles, aziridines, phthalazinones) in the context of diversity-oriented synthesis. This methodology was applied in over fifty examples and was used for the synthesis of five compounds of pharmaceutical interest. Of note, diversification of diaziridines into amines, hydrazines, and N-heterocycles had already been investigated by the authors, as noted in their earlier publication (Ref 18, *J. Am. Chem. Soc.*, 142, 21743-21750 (2020)). Finally, a route to ¹⁵N-labelled trifluorophenyl diazirine was developed for the preparation of labelled RNA splicing modulator candidates

- This article highlights a significant number of important points and undoubtedly represents excellent research work carried out by the authors. They thoroughly investigated this new hydroamination reaction, applying it to numerous examples, including terpene natural products and pharmaceuticals. This methodology proved to be tolerant of a wide range of functional groups, such as epoxides, aldehydes, and ketones. This article meets the standards of Nature, however the following issues should be raised

- It appears that the major concern of this article is that this methodology might result in the formation of racemic mixtures of amines and hydrazines. In fact, no enantioselectivity is mentioned despite the use of a chiral Cobalt-based catalyst. Has the author investigated the enantioselectivity of the corresponding amines or hydrazines? This point is crucial, particularly when the author claims access to pharmaceuticals, mostly requiring high enantiopurities.

- A number of other relevant studies on hydroamination and hydrohydrazination related to this manuscript or complementary strategy have been reported and could also be cited, such as the work by R. Knowles et al. (*J. Am. Chem. Soc.*, 2019, 141, 16590–16594) and J. Hartwig Ma et al. (*Chem*, 8, 532–542). The article by Knowles is particularly interesting because it describes the introduction of a wide scope of primary amines into unactivated olefins through a one-step process. In Hartwig's study, the amine-precursor is introduced enantioselectively.

Besides, Brunner pioneered the use of trifluoromethylphenyl diazirine for photolabelling studies, this article could also be cited: *J. Biol. Chem.* 1980, 3313.

- Why did the authors not work on a larger scale (than 0.1 mmol) to obtain more reliable isolated yields especially for hydrodiazirination reactions starting from readily available or commercial olefins (since the diazirine is accessible to the gram-scale) ?

Typos:

- Page 4: there an extra space below Fig 5

- There is a problem with the molecules numbering in Figure 5; they do not always correspond to those in the text. For example, phthalazinone is numbered as 25 or 26, and neranexane as 10 or 12, quinocide 8 or 12 etc

Reviewer #2 (Remarks to the Author):

In this manuscript, Lopchuk and coworkers reported a Markovnikov radical hydroamination

of unactivated alkenes with exclusive regioselectivity by Co-HAT strategy. Over fifty products with diverse functional groups were obtained in satisfactory yield, and the beautiful synthetic applications highlighted the practicability of this protocol. Particularly, the designed synthesis of ^{15}N -diazirine and its application in the preparation of ^{15}N -containing drug candidate is very impressive. However, after carefully reviewing the full manuscript, I found that this work does not meet the requirement of novelty to warrant publication as a paper in Nature Communications. The cobalt-catalyzed hydrogen atom transfer (HAT) strategy, which was employed in this work, for the selective hydroamination of unactivated alkenes has been reported by many groups (ACS Catal. 2020, 10, 4983; Angew. Chem. Int. Ed. 2021, 60, 25949; Nat. Commun. 2021, 12, 2552; Org. Lett. 2022, 24, 22; Angew. Chem. Int. Ed. 2023, 62, e202213913). Moreover, the use of diazirine for the formation of C-N bonds and subsequent derivatization of the products have been reported in previous works by the same group (J. Am. Chem. Soc. 2020, 142, 21743; Org. Lett. 2021, 23, 8838). Compared with their previous works, it did not seem to exhibit additional innovation in this work. Therefore, I don't recommended it to publish in Nature Communications.

Reviewer #3 (Remarks to the Author):

The authors described a regioselective hydroamination of unactivated olefins. The diazirines were used as the diversifiable nitrogen source.

This tactic exhibits a broad functional group tolerance. Notably, the terpenoid natural products could also be subjected to the standard reaction conditions. Furthermore, the illustrated fruitful, efficient synthesis of valuable compounds including clinical candidates and isotopic labeling molecules in this paper reveals the practicability and potential interest of this protocol.

Nevertheless, there are several issues that should be noted.

In the Figure 5C and Figure 6E, some assigned compound numbers in the descriptions on the synthesis of phthalazinone 26 and isotopic labeling ^{15}N -25 do not match with the charts. Please check the assigned numbers carefully.

The enantiopure chiral Co catalyst was used in the investigations. Have the authors detected the enantioselectivity of this regioselective hydroamination?

If there is no control on the enantioselectivity, is it necessary to use the optically pure Co catalyst?

The HRMS analysis is not within the 10 ppm acceptable error for several compounds: 3b, 3j, 3ac, 4d and 32.

The lacking of HRMS analysis for compound 4a should be noted.

Response to Reviewer Comments: NCOMMS-24-23795-T

Reviewer #1 (Remarks to the Author):

Comment: It appears that the major concern of this article is that this methodology might result in the formation of racemic mixtures of amines and hydrazines. In fact, no enantioselectivity is mentioned despite the use of a chiral Cobalt-based catalyst. Has the author investigated the enantioselectivity of the corresponding amines or hydrazines? This point is crucial, particularly when the author claims access to pharmaceuticals, mostly requiring high enantiopurities.

Response: The reactions are not asymmetric. We investigated the enantioselectivity, but racemic mixtures are obtained. In the SI (page S38), it is shown that both enantiomers of the catalyst performed the same.

Comment: A number of other relevant studies on hydroamination and hydrohydrazination related to this manuscript or complementary strategy have been reported and could also be cited, such as the work by R. Knowles et al. (J. Am. Chem. Soc., 2019, 141, 16590–16594) and J. Hartwig Ma et al. (Chem, 8, 532–542). The article by Knowles is particularly interesting because it describes the introduction of a wide scope of primary amines into unactivated olefins through a one-step process. In Hartwig's study, the amine-precursor is introduced enantioselectively. Besides, Brunner pioneered the use of trifluoromethylphenyl diazirine for photolabelling studies, this article could also be cited: J. Biol. Chem. 1980, 3313.

Response: The three references were added as 4, 5 and 23, and the reference list was updated.

Comment: Why did the authors not work on a larger scale (than 0.1 mmol) to obtain more reliable isolated yields especially for hydrodiazirination reactions starting from readily available or commercial olefins (since the diazirine is accessible to the gram-scale) ?

*Response: We feel this is a fairly standard reaction scale on which to operate for the majority of the scope table. For most of our examples, the resulting diaziridines are obtained in amounts from 20 to 40 mg, which is more than sufficient to calculate reliable isolated yields. Certain volatile starting materials (e.g. [1.1.1]propellane (**5e**), norborene (**5c**)), were conducted on 0.2 mmol scale. Furthermore, the reaction was demonstrated on 1 mmol scale for diaziridines **3o** and **3p** (~350-400 mg of product) and on gram scale for diaziridine **3h**.*

Comment: Page 4: there an extra space below Fig 5

Response: The space below Fig 5 was deleted.

Comment: There is a problem with the molecules numbering in Figure 5; they do not always correspond to those in the text. For example, phthalazinone is numbered as 25 or 26, and neranexane as 10 or 12, quinocide 8 or 12 etc

Response. We appreciate the reviewer catching this. The numbering has been corrected. “26” was changed to “25”, “quinocide 12” changed to “quinocide 8”, “mecamylamine 8” was changed to “mecamylamine 10” and “neramexane•HCl (10)” was changed to “neramexane•HCl (12)”.

Reviewer #2 (Remarks to the Author):

Comment: In this manuscript, Lopchuk and coworkers reported a Markovnikov radical hydroamination of unactivated alkenes with exclusive regioselectivity by Co-HAT strategy. Over fifty products with diverse functional groups were obtained in satisfactory yield, and the beautiful synthetic applications highlighted the practicability of this protocol. Particularly, the designed synthesis of ¹⁵N-diazirine and its application in the preparation of ¹⁵N-containing drug candidate is very impressive. However, after carefully reviewing the full manuscript, I found that this work does not meet the requirement of novelty to warrant publication as a paper in Nature Communications. The cobalt-catalyzed hydrogen atom transfer (HAT) strategy, which was employed in this work, for the selective hydroamination of unactivated alkenes has been reported by many groups (ACS Catal. 2020, 10, 4983; Angew. Chem. Int. Ed. 2021, 60, 25949; Nat. Commun. 2021, 12, 2552; Org. Lett. 2022, 24, 22; Angew. Chem. Int. Ed. 2023, 62, e202213913). Moreover, the use of diazirine for the formation of C-N bonds and subsequent derivatization of the products have been reported in previous works by the same group (J. Am. Chem. Soc. 2020, 142, 21743; Org. Lett. 2021, 23, 8838). Compared with their previous works, it did not seem to exhibit additional innovation in this work. Therefore, I don't recommend it to publish in Nature Communications.

Response: The reviewer correctly indicates that our work utilizes a Co-HAT hydroamination strategy, and that our regioselective hydroamination reaction shares a similar mechanism to other reactions in the literature, as we also described in our introduction. Neither the conception nor execution of this work was intended to develop a new reaction mechanism. However, with respect to the specific references the reviewer points to, the majority (e.g. ACS Catal. 2020, 10, 4983; Angew. Chem. Int. Ed. 2021, 60, 25949; Nat. Commun. 2021, 12, 2552; Angew. Chem. Int. Ed. 2023, 62, e202213913) pair an electrophilic Co(IV) center with a nucleophilic amine source, while our reaction does the exact opposite (nucleophilic radical carbon center with an electrophilic amine source). Ye's work (Org. Lett. 2022, 24, 22) is mechanistically more similar than the others but results in benzenesulfonimides, which are difficult to further manipulate and have not been demonstrated to be diversified.

*Overall, the novelty of this manuscript stems from several areas: 1) the first reported use of diazirine **1** as an electrophilic nitrogen source in a hydroamination reaction; (2) new strategic C-N bond disconnections from a highly diversifiable diaziridine intermediate that allows for the late-stage amination of natural products (Fig. 4), target-oriented synthesis of several pharmaceuticals (Fig. 5A/C), diversity-oriented synthesis of pharmaceutical intermediates, facilitating efficient medicinal chemistry workflows (Fig. 5B), and the late-stage amination of pharmaceutical candidates (Fig. 5C); and 3) a novel synthesis of bis-¹⁵N diazirine **1**, from a*

single source of the isotopic label (Fig. 6D), that was demonstrated in the preparation of RNA splicing modulator candidate **26** (Fig. 6E).

Our former work (*J. Am. Chem. Soc.* 2020, 142, 21743; *Org. Lett.* 2021, 23, 8838.) describes the development of the decarboxylative amination of redox-active esters, while this manuscript reports the regioselective hydroamination of unactivated alkenes. These are entirely different reactions; the current hydroamination now allows for the use of ubiquitous olefins, the targeting and exploration of different chemical space, and the deployment of different retrosynthetic disconnections.

Reviewer #3 (Remarks to the Author):

Comment: In the Figure 5C and Figure 6E, some assigned compound numbers in the descriptions on the synthesis of phthalazinone 26 and isotopic labeling 15N-25 do not match with the charts. Please check the assigned numbers carefully.

Response: We appreciate the reviewer catching this. The numbering has been corrected as described above for Reviewer #1.

Comment: The enantiopure chiral Co catalyst was used in the investigations. Have the authors detected the enantioselectivity of this regioselective hydroamination?

Response: The reactions are not asymmetric. We investigated the enantioselectivity, but racemic mixtures are obtained. In the SI (page S38), it is shown that both enantiomers of the catalyst performed the same.

Comment: If there is no control on the enantioselectivity, is it necessary to use the optically pure Co catalyst?

Response: No, the use of optically pure catalyst is not necessary. After determination that the enantiomer of the catalyst was of no consequence in the reaction (see above response), the optically pure catalyst was used just as a matter of practical availability in the lab.

Comment: The HRMS analysis is not within the 10 ppm acceptable error for several compounds: 3b, 3j, 3ac, 4d and 32. The lacking of HRMS analysis for compound 4a should be noted.

*Response: The HRMS data was recollected after re-tuning of the instrument and is now within 10 ppm. The HRMS for **4a** was added.*